# Retusone A, a Guaiane-Type Sesquiterpene Dimer from *Wikstroemia retusa* and Its Inhibitory Effects on Histone Acetyltransferase HBO1 Expression

**DOI:** 10.3390/molecules27092909

**Published:** 2022-05-03

**Authors:** Young Sook Yun, Tomomi Nakano, Haruhiko Fukaya, Yukio Hitotsuyanagi, Miho Nakamura, Megumi Umetsu, Nobuko Matsushita, Katsunori Miyake, Hiroyuki Fuchino, Nobuo Kawahara, Fuki Moriya, Akihiro Ito, Yuji Takahashi, Hideshi Inoue

**Affiliations:** 1School of Life Sciences, Tokyo University of Pharmacy and Life Sciences, 1432-1 Horinouchi, Tokyo 192-0392, Japan; tmpy10821082@gmail.com (T.N.); fukayah@toyaku.ac.jp (H.F.); yukioh@toyaku.ac.jp (Y.H.); nanaon2523zona@gmail.com (M.N.); nstawn.1wn6aw@gmail.com (M.U.); matsun@toyaku.ac.jp (N.M.); miyake@toyaku.ac.jp (K.M.); s167059@toyaku.ac.jp (F.M.); aito@toyaku.ac.jp (A.I.); yuji@toyaku.ac.jp (Y.T.); hinoue@toyaku.ac.jp (H.I.); 2Research Center for Medicinal Plant Resources, National Institutes of Biomedical Innovation, Health and Nutrition, Tsukuba 305-0843, Japan; fuchino@nibiohn.go.jp (H.F.); kawahara@nibiohn.go.jp (N.K.)

**Keywords:** *Wikstroemia retusa*, Thymelaeaceae, sesquiterpene, guaiane-type, HBO1, breast cancer

## Abstract

Retusone A (**1**), a new sesquiterpene dimer consisting of two guaiane-type sesquiterpenoids, and oleodaphnal (**2**) were isolated from heartwood of *Wikstroemia retusa* (Thymelaeaceae). The planar structure of **1** was elucidated on the basis of HRESIMS and NMR spectroscopic data, and the relative stereochemistry was established by X-ray diffraction analysis. The absolute configuration of **1** was determined by electronic circular dichroism. Compound **1** suppressed luciferase reporter gene expression driven by the HBO1 (histone acetyltransferase binding to ORC1) gene promoter in human breast cancer MCF7 cells. Compound **1** also decreased the expression of endogenous HBO1 mRNA and protein, and inhibited proliferation of the cells. These results suggest that retusone A (**1**), which has a unique dimeric sesquiterpenoid structure with inhibitory activity against HBO1 expression, may contribute to the development of a novel therapeutic candidate for the treatment of breast cancer.

## 1. Introduction

Breast cancer predominantly affects women and is the most common cancer among women worldwide, with an estimated 2 million cases reported each year [1]. HBO1, a histone acetyltransferase that binds to the origin recognition complex 1 (ORC1) [2], is highly expressed in carcinomas of the testis, ovary, breast, stomach/esophagus, and bladder [3], and is known to promote cell proliferation in bladder and breast cancers [4]. HBO1 is essential for replicational licensing [5] medicated by histone 3 and 4 acetylation and is involved in embryonic development and the survival of erythroblasts in the fetal liver [6,7]. Thus, the development of HBO1-targeting molecules could contribute to our understanding of the mechanism of HBO1 regulation and could have potential therapeutic applications.

*Wikstroemia retusa* A. Gray is endemic to the Ryukyu Islands, Japan, and is used locally as a source of paper. The genus Wikstroemia (Thymelaeaceae) belongs about 70 species widely distributed in East Asia, Malaysia, Australia, and China [8]. Several guaiane-type sesquiterpenoids have been reported from *W*. *lanceolate* [9], *W*. *scytophylla* [10], and *W*. *indica* [11]. However, the only chemical constituents of this plant reported to date are daphnane-type diterpenoids [12,13,14] and a coumarin daphnoretin [15].

By screening a methanol plant extract library based on an HBO1 promoter-driven reporter gene assay, an extract of *W*. *retusa* heartwood was found to suppress HBO1 promoter-dependent transcription without showing cytotoxic effects on human breast cancer MCF7 cells. Here, we report the isolation and structure determination of a new dimeric sesquiterpenoid as an active compound from this plant, and its effects on endogenous HBO1 expression and inhibition of MCF7 cell proliferation.

## 2. Results

### 2.1. Structure Elucidation of ***1*** and ***2***

The methanol extract (348.6 g) of *W*. *retusa* heartwood (dry weight 3445 g) was dried, then dissolved in methanol, and the methanol layer and insoluble material (83.6 g) were separated by decantation. After concentration, the separated methanol layer was partitioned between ethyl acetate and water to obtain ethyl acetate-soluble material (39.8 g). The methanol-insoluble part obtained from the methanol extract was chromatographed on a silica gel column and eluted with a hexane/ethyl acetate mixture and then ethyl acetate/methanol mixtures to give nine fractions (Frs. 1–9). Fractions 4–6 showed inhibitory activity against HBO1 promoter-driven reporter expression. Fraction 4 was separated by HPLC to give compound **1** (4.0 mg) (Figure 1). Compound **2** (16.7 mg) was obtained by chromatographic separation of Fr. 6 and the ethyl acetate-soluble material and identified as oleodaphnal by comparison of its NMR and MS spectra with those in the literature [16] (Figure 1 and Appendix A, see Appendix A).

Compound **1** was obtained as colorless plates, and its molecular formula was established as C_31_H_38_O_4_ by HRESIMS measurements ([M + Na]^+^ *m*/*z* 497.2663, calcd for C_31_H_38_O_4_Na 497.2668), indicating 13 degrees of unsaturation. Its IR spectrum displayed the characteristic absorptions of carbonyl (1697 cm^−1^) groups. The ^1^H NMR spectrum of **1** (Table 1) exhibited signals for four methyl groups (*δ*_H_ 1.74, 1.78, 1.79, and 1.80), six downfield-resonated protons (*δ*_H_ 4.40, 4.71, 4.75, 4.77, 4.81, and 4.86), and one methoxy group (*δ*_H_ 3.39). The ^13^C NMR (Table 1), DPET, and HMQC spectra (Appendix A, see Appendix A) indicated the presence of 31 carbon signals, including four methyl carbons (*δ*_C_ 8.3, 8.4, 20.6, and 20.7), seven aliphatic methylene carbons (*δ*_C_ 24.3, 29.4, 31.9, 32.3, 32.5, 35.7, and 39.1), three aliphatic methine carbons (*δ*_C_ 42.7, 42.9, and 47.7), one oxymethine carbon (*δ*_C_ 69.8), one acetal carbon (*δ*_C_ 99.2), two olefinic methylene carbons (*δ*_C_ 109.7 and 110.2), ten olefinic quaternary carbons (*δ*_C_ 130.7, 133.8, 135.9, 136.6, 140.0, 140.2, 148.7, 149.4, 167.4, and 168.3), two carbonyl carbons (*δ*_C_ 200.1 and 204.6), and one methoxy carbon (*δ*_C_ 55.7). Six double bonds and two carbonyls accounted for eight degrees of unsaturation, and the remaining five degrees of unsaturation required **1** to be hexatacyclic. In unit I of **1**, the HMBC (Appendix A, see Appendix A correlations of H-2/C-1 and C-3, H_2_-6/C-1 and C-4, H_2_-8/C-6 and C-10, H_2_-9/C-1 and C-7, and H_3_-15/C-3 and C-5 indicated a 3-oxo-4-methylbicyclo[5.3.0]decane carbon framework. The HMBC correlations of H_2_-12/C-7 and C-13, and H_3_-13/C-7 and C-12 suggested the presence of an isopropenyl group at C-7. Therefore, the structure of unit I was confirmed as a guaiane-type sesquiterpenoid. As shown in Figure 2a, the HMBC correlations of unit II of **1** are similar to those of unit I, indicating that the structure of unit II is also a guaiane-type sesquiterpenoid. These HMBC spectrum of **1** revealed that **1** was composed of two guaiane units (I and II; Figure 2a). The observed ^1^H-^1^H COSY (Appendix A, see Appendix A) correlation between H-2 (*δ*_H_ 3.09) and H-14′ (*δ*_H_ 4.40), HMBC correlations from H-2 to C-10′ (*δ*_C_ 135.9) and C-14′ (*δ*_C_ 69.8), and from H-14′ to C-1 (*δ*_C_ 133.8) and C-3 (*δ*_C_ 200.1) indicated that C-2 of unit I was connected to C-14′ of unit II. C-14 of unit I was an acetal carbon, to which H-14′ of unit II and the methoxy protons showed HMBC correlations, suggesting that C-1, C-2, C-10, C-14, C-14′, and an oxygen atom form a dihydropyran ring and a methoxy group attached to C-14. From these observations, the planar structure of compound **1** was elucidated as shown in Figure 2a. Careful crystallization of **1** from methanol gave crystals suitable for X-ray diffraction analysis, which revealed the relative stereochemistry of **1** as shown in Figure 2b (Appendix A, see Appendix A).

To establish the absolute configuration of **1**, the electronic circular dichroism (ECD) spectrum of **1** was compared with the time-dependent density functional theory (TDDFT) calculated spectrum of (2*S*,7*R*,14*S*,7′*R*,14′*S*)-**1** at the B3LYP/6–311+G (2df, 2p) level. The calculated ECD spectrum of **1** was very similar to the experimental plot of **1**, supporting that **1** has a 2*S*,7*R*,14*S*,7′*R*,14′*S* configuration (Figure 3 and Appendix A, and Appendix A, see Appendix A). Thus, the structure of **1** was determined as shown in Figure 1 and named retusone A.

The two guaiane units of **1** are structurally similar to oleodaphnal (**2**), which was also isolated in this study. This suggests that **1** is biosynthesized from **2**. A possible biosynthetic pathway for **1** from two molecules of **2** is shown in Figure 4. A coupling product of two molecules of **2** is generated by an aldol-type reaction between the C-2 methylene carbon atom of one and the C-14 carbonyl carbon of the other, and it is readily converted into a hemiacetal structure at C-14. The *O*-methylation of the hemiacetal hydroxy group or substitution of the hydroxy group by a methoxy group gives **1**.

### 2.2. Inhibitory Effects of ***1*** on HBO1 Expression and Cell Proliferation

MCF-HBO1Luc cells were created by stable transfection of MCF7 cells with a human HBO1-Luc construct. The cells were treated with increasing concentrations of **1** for 24 h; luciferase activity was reduced by about 50 and 70% at 5 and 10 µM, respectively, compared to the control (Figure 5a). The WST-8 assay was performed to confirm that the decrease in luciferase activity was not due to reduced cell viability. Treatment with 10 µM of **1** for 24 h did not alter cell viability compared to the control, while 10 µM etoposide decreased cell viability (Figure 5b). To investigate the effects of **1** on the expressions of endogenous HBO1, mRNA and protein levels were evaluated by RT-qPCR and Western blotting, respectively. Treatment with **1** (10 µM) for 24 h significantly reduced both mRNA and protein levels by about 30% compared to the control (Figure 5c,d).

The long-term effect of **1** on MCF7 proliferation was evaluated by the colony formation assay. After treating the cultured MCF7 cells with **1** (10 μM) for 14 days, a decrease in the number of MCF7 colonies was observed when assessed by the crystal violet staining method (Figure 6) [17].

## 3. Discussion

In this study, we isolated a monomeric sesquiterpenoid, oleodaphnal, and a new dimeric sesquiterpenoid retusone A (**1**) from *W. retusa*. Although oleodaphnal has been reported from *W. indica* [18] and *W. coriacea* [19], this is the first report from *W. retusa*. Except for its chemical structure, very little is known about the biological activity of oleodaphnal. Oleodaphnal did not express luciferase reporter activity (data not shown). However, retusone A (**1**), the dimer of oleodaphnal, reduced cell proliferation and decreased the expression of the HBO1 promoter, mRNA, and protein. These results suggest that the activity of retusone A (**1**) is manifested after the dimerization of two uniquely bridged hexacyclic structures of two oleodaphnal monomers. 

A synthetic compound, WM-3835, was recently reported to inhibit HBO1 activity in leukemia stem cells. WM-3835 contains an acylsulfonylhydrazide moiety and binds directly to the acetyl CoA binding site of HBO1[20]. It is the only known small molecule inhibitor of HBO1. Reduced HBO1 expression has been implicated in cancer cell proliferation. For example, the HBO1 shRNA inhibits the viability, proliferation, and migration of hepatocellular carcinoma cells [21]. Moreover, HBO1 depletion inhibits the growth of anti-estrogen-treated breast cancer cells [22]. The inhibition of cell proliferation by retusone A (**1**) is presumed to be directly related to inhibiting HBO1 expression. Breast cancer is the most common cancer among women, and its incidence is expected to increase continually. Therefore, further efforts should be made to identify the correlation between the suppression of cell proliferation by retusone A (**1**) and the suppression of HBO1 expression.

## 4. Materials and Methods

### 4.1. General Experimental Procedures

Melting points were determined on a Yanaco MP-3 apparatus (Anatec Yanaco Corp., Kyoto, Japan). Optical rotation was measured on a JASCO DIP-1000 digital polarimeter (JASCO Corp., Tokyo, Japan). IR spectra were recorded on a JASCO FT-IR 620 spectrophotometer (JASCO Corp., Tokyo, Japan). UV spectra were obtained using a Hitachi U-2001 spectrophotometer (Hitachi High-Tech Corp., Tokyo, Japan). NMR spectra were measured on a Bruker Ascend 500 spectrometer (Bruker Corp., Billerica, MA, USA) at 300 K. The ^1^H NMR chemical shifts in CDCl_3_ were calibrated to the residual CHCl_3_ resonance at 7.26 ppm, and the ^13^C NMR chemical shifts were calibrated to the solvent peak at 77.0 ppm. Mass spectra were obtained using a Waters Xevo G2-XS QTof mass spectrometer (Waters Corp., Milford, MA, USA). CD spectra were measured on a JASCO J-1500 CD spectrometer (JASCO Corp., Tokyo, Japan). Preparative HPLC was carried out on a Shimadzu LC-10AT system equipped with an SPD-10AVP detector (Shimadzu Corp. Kyoto, Japna) and a Mightysil RP-18 prep column (5 μm, 20 × 250 mm). All of the organic solvents were purchased from Kanto Chemicals (Japan). 

### 4.2. Plant Materials

*W*. *retusa* was cultivated in the greenhouse of Tokyo University of Pharmacy and Life Sciences. The origin of the plant was identified by Dr. Katsunori Miyake, and a voucher specimen (Wr-2019) has been deposited in the Laboratory of Molecular and Chemical Biology, School of Life Sciences, Tokyo University of Pharmacy and Life Sciences.

### 4.3. Extraction and Isolation of Compounds

Dried *W. retusa* heartwood (3445 g) was extracted with methanol (3 × 25 L) at room temperature. After filtration, evaporation of the solvent under reduced pressure gave a methanolic extract (348.6 g). Methanol (300 mL) was added to the extract, and the mixture was stirred vigorously. The insoluble material (83.6 g) was separated by decantation, and after concentration under reduced pressure, the methanol-soluble portion was partitioned between ethyl acetate and water. The ethyl acetate layer was concentrated under reduced pressure to give ethyl acetate-soluble material (39.8 g). The methanol-insoluble part (83.6 g) was loaded onto a silica gel column and eluted sequentially with *n*-hexane/ethyl acetate mixtures (20:1, 10:1, 8:2, 7:3, 6:4, 1:1, 3:7, and 0:1, *v*/*v*) and then ethyl acetate/methanol mixtures (9:1, 8:2, 7:3, 6:4, and 1:1, *v*/*v*) to give nine fractions: Fr. 1 (0.4 g), Fr. 2 (0.7 g), Fr. 3 (0.6 g), Fr. 4 (3.7 g), Fr. 5 (1.6 g), Fr. 6 (11.6 g), Fr. 7 (7.3 g), Fr. 8 (28.8 g), and Fr. 9 (8.1 g). Fraction 4 (3.7 g), obtained by elution with *n*-hexane/ethyl acetate (6:4 and 1:1, *v*/*v*), showed HBO1 promoter inhibition activity of about 60% at 10 µg/mL and was further separated by preparative HPLC with a mixture of H_2_O/acetonitrile (20:80, *v*/*v*) to give compound **1** (4.0 mg). Fraction 6 was further fractionated by octadecylsilyl-silica gel column chromatography with methanol/water (1:1, 4:1, and 1:1, *v*/*v*) to give three sub-fractions: Fr. 6-1 (1.06 g), Fr. 6-2 (7.69 g), and Fr. 6-3 (2.62 g). Compound **2** (9.4 mg) was obtained from Fr. 6-2 by HPLC with water/acetonitrile (35:65, *v*/*v*).

The ethyl acetate-soluble material was loaded onto a silica gel column and eluted with *n*-hexane/ethyl acetate mixtures (1:1 and 0:1, *v*/*v*) and then ethyl acetate/methanol mixtures (10:1, 9:1, 8:2, 7:3, 6:4, and 1:1, *v*/*v*) to give five fractions: Fr. E1 (0.2 g), Fr. E2 (1.5 g), Fr. E3 (24.9 g), Fr. E4 (6.3 g), and Fr. E5 (3.5 g). Fraction E3 was subjected to octadecylsilyl-silica gel chromatography with methanol/water (1:1, 4:1, and 1:0, *v*/*v*) to afford three sub-fractions: Fr. E3-1 (10.3 g), Fr. E3-2 (13.2 g), and Fr. E3-3 (1.1 g). Fraction E3-3 was separated by HPLC using water/acetonitrile (45:55, *v*/*v*) to give compound **2** (7.3 mg).

### 4.4. Structural Elucidation

*Retusone A* (**1**): colorless plates; mp 166–168 °C; [α]_D_^25^ + 282 (*c* 0.08, MeOH); UV (MeOH) λmax (log *ε*) 290 (4.04) nm (Appendix A, see Appendix A; ECD (MeOH) λmax (Δ*ε*, 0.04 mg/mL) 287 (−2.46), 337 (+1.71) nm; IR (film) ν_max_ 2922 (CH), 1697 (C = O) cm^−1^, Appendix A, see Appendix A); ^1^H NMR and ^13^C NMR data, Table 1 (Appendix A, see Appendix A); HRESIMS *m*/*z* 497.2663 [M+Na]^+^ (calcd for C_31_H_38_O_4_Na 497.2668, Appendix A, see Appendix A).

### 4.5. Crystallography

CCDC 2078448 contains the supplementary crystallographic data for **1**. A copy of the data can be obtained free of charge on application to the CCDC, 12 Union Road, Cambridge, CB2 IEZ, UK; fax: +44 (0) 1223 336033; or e-mail: deposit@ccdc.cam.ac.uk.

### 4.6. Calculation of the ECD Spectrum

A Monte Carlo (MC) conformational search of **1** with a 2*S*,7*R*,14*S*,7′*R*,14′*S* configuration was configured using the “automatic setup” routine in MacroModel [23]. The calculation consisted of 50,000 MC steps with 500 iterations per step using the MMFF94s force field and the PR conjugate gradient with no solvation. The obtained 30 conformers within 4 kcal/mol were subject to density functional theory (DFT) calculations [24] at the B3LYP/6–311+G (2df, 2p) level in MeOH to obtain 5 conformers for **1** within 2 kcal/mol from the global minimum energy conformation. The ECD spectra calculations were performed for these 5 conformers at the B3LYP/6–311+G (2df, 2p) level in MeOH. The calculated relative energies and the Boltzmann populations are summarized in Appendix A. The theoretical ECD spectrum of **1** was obtained by producing the spectra of those conformers weighted according to the Boltzmann population [25].

### 4.7. Cell Culture

The human breast cancer cell line MCF7 was cultured in Dulbecco’s Modified Eagle Medium (DMEM) with 10% fetal bovine serum and 2 mM glutamine at 37 °C under 5% CO_2_.

### 4.8. In Vitro Reporter Gene Assay

To construct human HBO1-Luc, a 640 -bp (-640/-1) fragment of the HBO1 gene promoter was cloned in front of the luciferase reporter. We generated a MCF7 cell line with stable expression of human HBO1-Luc, called MCF-HBO1Luc cells. The cells were seeded in 96-well plates at a density of 1.0 × 10^4^ cells per well and incubated at 37 °C. After 24 h, compound **1** was added to the cells and incubated at 37 °C for 24 h. The cells were washed with PBS and lysed with Bright-Glo™ Reagent (Promega Co., Madison, WI, USA), and luciferase activity in the cell lysates was determined using a multimode plate reader (EnSpire, Perkin-Elmer, Waltham, MA, USA). 

### 4.9. Determination of Cell Viability

The WST-8 assay using a Cell Counting Kit-8 (Dojin, Kumamoto, Japan) was used to determine cell viability following the manufacturer’s instructions. Cells were seeded in 96-well plates, cultured overnight, and treated with compound **1** or etoposide (10 μM) as a positive control. After adding WST-8, the cells were incubated for 4 h, and the absorbance was measured at 450 nm using an EnSpire multimode plate reader.

### 4.10. The Quantitative Reverse Transcription Polymerase Chain Reaction (qRT-qPCR)

Total RNA was extracted using a FavorPrep Tissue Total RNA Mini Kit (Favorgen, Pingtung, Taiwan), and total RNA was then reverse transcribed using ReverTra Ace qPCR RT Master Mix (TOYOBO, Osaka, Japan). Subsequently, RT-qPCR analysis was performed according to the manufacturer’s recommendations (denaturation at 95 °C for 1 min, followed by 45 cycles of 95 °C for 10 s, 60 °C for 30 s) in a CFX96 Real-Time PCR Detection System (BIO RAD, Hercules, CA, USA).

The primer sequences for HBO1 were as follows: forward, 5′-TCTCCGCTACCTGCATAATTTTCAAGGC-3′, and reverse, 5′-TTGGAGTTGGACCTTTTGGCCTCTTTGG-3′. For GAPDH, they were as follows: forward, 5′-GCACCGTCAAGGCTGAGAAC-3′, and reverse, 5′-TGGTGAAGACGCCAGTGGA-3′. All of the experiments were repeated three times.

### 4.11. Western Blot Analysis

Cell extracts were prepared in RIPA buffer (Fuji Film, Tokyo, Japan) containing 1× cOmplete^TM^ protease inhibitor cocktail (Roche Diagnostics, Basel, Switzerland). The protein concentration was determined using a BCA protein assay kit (Thermo Fisher Scientific, Waltham, MA, USA). Lysates were heated at 95 °C for 5 min, subjected to SDS-PAGE, transferred to polyvinylidene difluoride transfer membranes, incubated with primary and secondary antibodies, and visualized using Immobilon Western chemiluminescent HRP substrate (Millipore, Burlington, MA, USA), according to the manufacturer’s instructions. Primary antibodies used were anti-HBO1 (1:1000; Cell Signaling Technology, Danvers, MA, USA) and anti-*β*-actin (1:1000; Sigma-Aldrich, Saint Louis, MO, USA). Secondary antibodies were anti-rabbit HRP-conjugated antibody (1:1000; Santa Cruz Biotechnology, Dallas, TX, USA).

### 4.12. Crystal Violet Staining

MCF7 cells were seeded in 24-well plates at a density of 500 cells/well and incubated with compound **1** (10 µM) or etoposide (10 µM) for 14 days, during which the medium was exchanged with fresh medium every three days. Then, the medium was removed, the cells were washed with PBS and then fixed with 4% paraformaldehyde for 15 min. Following removal of the fixing agent, the cells were stained with 0.5% crystal violet in aqueous 25% methanol for 10 min and washed with distilled water, and morphological images were obtained with a scanner. The stained cells were then lysed with MeOH and centrifuged, and 100 µL of the supernatant was measured at 540 nm using an EnSpire multimode plate reader.

### 4.13. Statistical Analysis

All the results are expressed as mean ± SD, calculated from the results of at least three independent experiments. Statistical significances were determined by one-way analysis of variance (ANOVA) followed by Dunnett’s multiple comparison test or *t*-test. *p* < 0.05 was considered statistically significant.

## 5. Conclusions

A new compound retusone A (**1**), a dimeric sesquiterpenoid, was isolated from *W. retusa*. Retusone A (**1**) reduced HBO1 expression and cell proliferation. Thus, inhibition of HBO1 by retusone A could provide a new insight into developing novel therapeutic agents for breast cancer treatment.

## Figures and Tables

**Figure 1 molecules-27-02909-f001:**
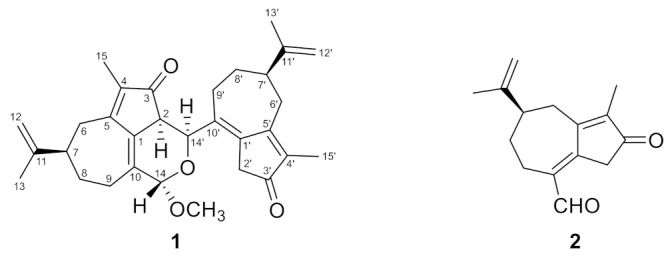
The structures of retusone A (**1**) and oleodaphnal (**2**).

**Figure 2 molecules-27-02909-f002:**
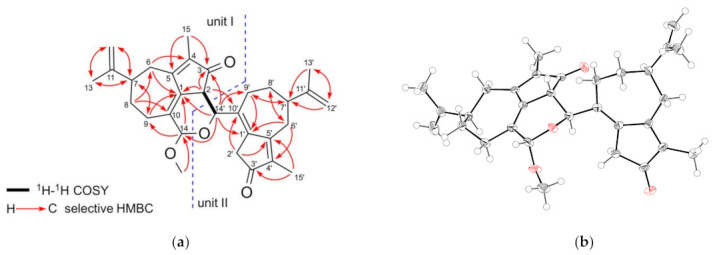
(**a**) Key HMBC and 1H-1H COSY correlations and (**b**) ORTEP drawing of the crystal structure of **1**.

**Figure 3 molecules-27-02909-f003:**
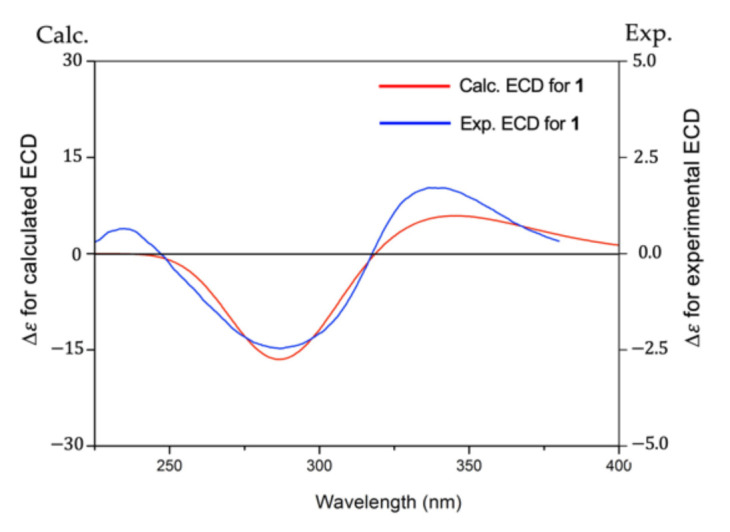
Calculated and experimental ECD spectra of **1**.

**Figure 4 molecules-27-02909-f004:**
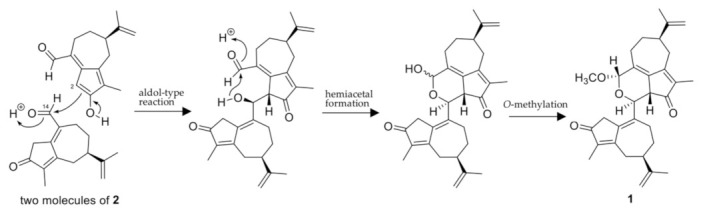
A possible biosynthetic pathway from oleodaphnal (**2**) to retusone A (**1**).

**Figure 5 molecules-27-02909-f005:**
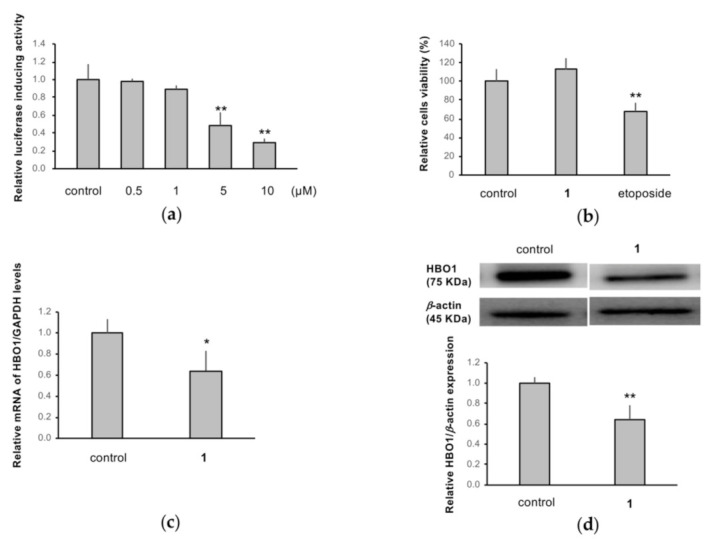
Inhibitory activities of retusone A (**1**) on HBO1 expression in MCF 7 cells. (**a**) Luciferase reporter expression from pHBO1p(-640/-1)-Luc, (**b**) cell viability, (**c**) mRNA, and (**d**) protein level of endogenous HBO1 after treatment with (0.5–10 µM) for (**a**) and 10 µM for (**b**–**d**) with or without **1**. Dimethylsulfoxide (DMSO) was used as a vehicle control. The quantitative data are shown as mean ± standard deviation (SD, *n* = 4). * *p* < 0.05 or ** *p* < 0.01 vs. control.

**Figure 6 molecules-27-02909-f006:**
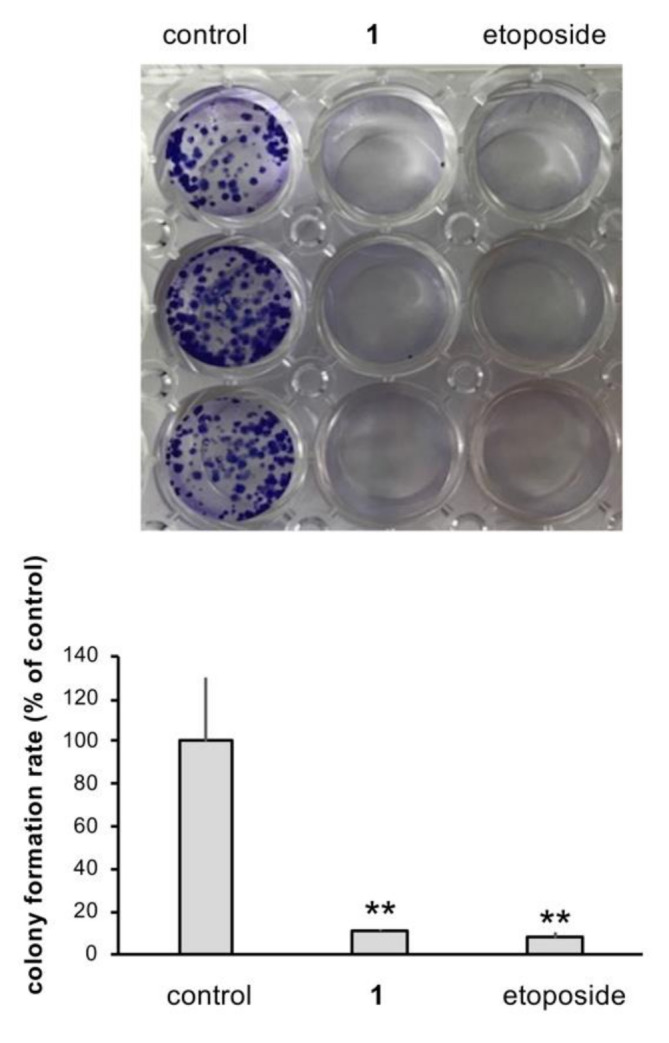
Colony formation assay on MCF7 cells treated with **1**. Dimethylsulfoxide (DMSO) was used as a vehicle control. The colony formation rate in the presence and absence of **1** or etoposide (10 μM) is shown as mean ± standard deviation (*n* = 3). ** *p* < 0.01 vs. control.

**Table 1 molecules-27-02909-t001:** 1H (500 MHz) and 13C (125 MHz) NMR Data of **1** in CDCl_3_
*^a^.*

Position	δ_H_	δ_C_ Type	COSY	HMBC
1		133.8, C		
2	3.09, d (10.4)	47.7, CH	14′	1, 3, 14′
3		200.1, C		
4		140.2, C		
5		163.8, C		
6	2.77, bd (17.8)	35.7, CH_2_	7	1, 5, 7
7	2.57 ^b^	42.9, CH	6, 8	9, 12, 13
8	1.87 ^b^, 2.00 ^b^	31.9, CH_2_	7, 9	6, 7, 9, 10
9	2.22, ddd (4.2, 9.4, 18.2), 2.63 ^b^	29.4, CH_2_	8	1, 7, 8
10		130.7, C		
11		148.7, C		
12	4.71, bm, 4.77, m	109.7, CH_2_	13	7, 11, 13
13	1.78, s	20.6, CH_3_	12	7, 11, 12
14	4.86, s	99.2, CH		1, 9, 10
15	1.74, s	8.3, CH_3_		3, 4, 5
1′		136.6, C		
2′	2.90 ^b,c^	39.1, CH		1′, 3′, 5′, 10
3′		204.6, C		
4′		140.0, C		
5′		167.4, C		
6′	2.84 ^b,c^	32.5, CH_2_	7′	1′, 4′, 5′, 7′, 8′
7′	2.60 ^b^	42.7, CH	6′, 8′	6′, 8′, 9′
8′	1.94, ddd (2.5, 7.1, 14.1), 2.02 ^b^	32.3, CH_2_	7′, 9′	1′, 4′, 5′, 7′, 8′
9′	2.50, ddd (2.5, 7.6, 16.4), 2.61 ^b^	24.3, CH_2_	8′	1′, 7′, 10′
10′		135.9, C		
11′		149.4, C		
12′	4.75, m, 4.81, bm	110.2, CH_2_	13′	7′, 11′, 13′
13′	1.79, s	20.7, CH_3_	12′	7′, 11′, 12′
14′	4.40, d (10.4)	69.8, CH	2′	1, 3, 14, 1′, 9′, 10′
15′	1.80, s	8.4, CH_3_		3′, 4′, 5′
OCH3	3.39, s	55.7, CH_3_		14

*^a^* J-values are given in Hz in parentheses. ^b^ Multiplicity not determined due to overlapping and/or broadening of the signals. ^c^ Two protons.

## Data Availability

Not applicable.

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
