# Peer review of "Retusone A, a Guaiane-Type Sesquiterpene Dimer from Wikstroemia retusa and Its Inhibitory Effects on Histone Acetyltransferase HBO1 Expression"

_molecules, 2022, doi:10.3390/molecules27092909_

Round 1

Reviewer 1 Report

The following MS cannot be accepted for publications in its current form. The bellow issues should be covered

Abstract

-Add the plant`s family name

-Add name of assay.

-Add plant fraction from which this compound separated.

Keywords

Should be more targeted, therefore, remove promoter; mRNA; protein; proliferation instead use breast cancer and plant`s family name.

Introduction

Are there any HBO1-targeting molecules reported from natural sources, especially from plant origin.

Is there any reported traditional uses of this plant, clarify?.

Authors should write about the genus, are there any guaiane-type sesquiterpenoids reported from the genus.

Results

-2.1. Structure Elucidation of 1 and 2

The first paragraph in this section should be summarized.  need to mention so much details in this section about the extraction and isolation of the compounds.

-Compound 2 should be added in the abstract.

-The COSY and HMBC correlations of the of units I and II should be discussed in details to prove the presence of each unit.

- The X-ray tool for configuration assignment should be added in the abstract.

- In table 1

1-multiplicity of the carbons as well as the COSY and HMBC correlations should be added.

-In figure 2a, it is so crowded, therefore remove the 2J HMBC corrections and keep only 3J HMBC correlations.

-Figure 4, it will be more significant if the enzymes responsible for each step in the biosynthesis as well as the reaction type to be added above the arrows.

Discussion

-It is missing.

-Are there any sesquiterpenes reported to have such activity??.

- Are the results obtained by authors in agreement with the reported results if present?.

Conclusions

-Is very weak needs to be rewritten.

Author Response

#1

Abstract

-Add the plant`s family name.

Response

We have added "(Thymelaeaceae) " to line 18 of abstract with accordance in referee’s comment.

-Add name of assay.

Response

In accordance with the referee's comments, we have added "luciferase" to line 21 of abstract.

-Add plant fraction from which this compound separated.

Response

We have described the isolation of compounds in detail in the Method & Materials section.

Keywords

-Should be more targeted, therefore, remove promoter; mRNA; protein; proliferation instead use breast cancer and plant`s family name.

Response

We have agreed with the referee’s comment. So, instead of "promoter; mRNA; protein; proliferation" we changed to" Thymelaeaceae; breast cancer" to line 28 of Keywords.

Introduction

-Are there any HBO1-targeting molecules reported from natural sources, especially from plant origin.

Response

There are few reports on natural compounds targeting HBO1.

-Is there any reported traditional uses of this plant, clarify?.

Response

W. retusa has been used locally as a source of pulp.

-Authors should write about the genus, are there any guaiane-type sesquiterpenoids reported from the genus.

Response

According to the comment of the referee, the following sentences were inserted in line 42-45.

"The genus Wikstroemia (Thymelaeaceae) belongs about 70 species widely distributed in East Asia, Malaysia, Australia, and China [8]. Several guaian-type sesquiterpenoids have been reported from W. lanceolate [9], W. scytophylla [10], and W. indica [11].

Results

-2.1. Structure Elucidation of 1 and 2

The first paragraph in this section should be summarized.  need to mention so much details in this section about the extraction and isolation of the compounds.

Response

We considered that the extraction and isolation of these two compounds were experimental operations and described in detail in the Methods and Materials section.

-Compound 2 should be added in the abstract.

Response

We have added "and oleodaphnal (2), were" to line 18 of abstract with accordance in referee’s comment.

-The COSY and HMBC correlations of the of units I and II should be discussed in details to prove the presence of each unit.

Response

Following the referee’s advices, the HMBC correlations for units I and II of compound 1 are detailed in lines 83-90.

- The X-ray tool for configuration assignment should be added in the abstract.

Response

X-ray diffraction analysis was used only as a confirmation of structure and relative arrangement. Therefore, I do not think it is necessary to add the configuration determination by X-ray diffraction to the abstract.

- In table 1

1-multiplicity of the carbons as well as the COSY and HMBC correlations should be added.

 Response

According to the comment of the referee, we have inserted carbons type, 1H-1H COSY and HMBC correlations in the Table 1.

-In figure 2a, it is so crowded, therefore remove the 2J HMBC corrections and keep only 3J HMBC correlations.

Response

We have deleted the 2J HMBC correlations in figure 2a except for H-2/C-1 and C-3, with accordance in referee’s advice.

-Figure 4, it will be more significant if the enzymes responsible for each step in the biosynthesis as well as the reaction type to be added above the arrows.

Response

We could not be described since enzyme has been not identified in the biosynthetic process from oleodaphnal to retusone A. So, we described only possible reactions in the biosynthetic process. However, we wrote reaction type on the arrows according to the comments of the referee.

Discussion

-It is missing.

Response

In response to comments from the referee, we have included the following text in the Discussion section (line 156-177).

"In this study, we isolated a monomeric sesquiterpenoid, oleodaphnal, and a new dimeric sesquiterpenoid retusone A (1) from W. retusa. Although oleodaphnal has been reported from W. indica and W. coriacea, this is the first report from W. retusa. Except for its chemical structure, very little is known about the biological activity of oleodaphnal. Oleodaphnal did not express luciferase reporter activity (data not shown). However, retusone A (1), the dimer of oleodaphnal, reduced cell proliferation and decreased the expression of the HBO1 promoter, mRNA, and protein. These results suggest that the activity of retusone A (1) is manifested after the dimerization of two uniquely bridged hexacyclic structures of two oleodaphnal monomers. A synthetic compound, WM-3835, was recently reported to inhibit the HBO1 activity in leukemia stem cells. WM-3835 contains an acylsulfonylhydrazide moiety and binds directly to the acetyl CoA binding site of HBO1. It is the only known small molecule inhibitor of HBO1. Reduced HBO1 expression and HBO1 activity have been implicated in cancer cell proliferation. For example, the HBO1 shRNA inhibits the viability, proliferation, and migration of hepatocellular carcinoma cells. Moreover, HBO1 depletion inhibits the growth of anti-estrogen-treated breast cancer cells. Inhibition of cell proliferation by retusone A (1) is presumed to be directly related to inhibiting HBO1 expression. Breast cancer is the most common cancer among women, and its incidence is expected to increase continually. Therefore, further efforts should be made to identify the correlation between the suppression of cell proliferation by retusone A (1) and the suppression of HBO1 expression. "

- Are the results obtained by authors in agreement with the reported results if present?.

Response

As far as we know, we have not found any reports similar to our results.

Conclusions

-Is very weak needs to be rewritten.

Response

We have changed the conclusion to the following sentence according to the referee's comment: "Retusone A (1), a novel dimeric sesquiterpenoid, was isolated from W retusa. We changed it to "A novel compound, retusone A (1), a dimeric sesquiterpenoid, was isolated from W retusa. retusone A (1) inhibited HBO1 expression and cell proliferation. Thus, inhibition of HBO1 by retusone A may provide new insights for the development of novel therapeutic agents for the treatment of breast cancer.

-Are there any sesquiterpenes reported to have such activity??.

Response

As far as we know, there are no inhibitors of HBO1 derived from natural products including sesquiterpenoids.

- Are the results obtained by authors in agreement with the reported results if present?.

Response

We have not found any reports similar to our results.

Conclusions

-Is very weak needs to be rewritten.

Response

We have changed the conclusion to the following sentence according to the referee's comment: "Retusone A (1), a novel dimeric sesquiterpenoid, was isolated from W retusa. We changed it to "A novel compound, retusone A (1), a dimeric sesquiterpenoid, was isolated from W retusa. retusone A (1) inhibited HBO1 expression and cell proliferation. Thus, inhibition of HBO1 by retusone A may provide new insights for the development of novel therapeutic agents for the treatment of breast cancer."

Reviewer 2 Report

The manuscript entitled (Retusone A, a Guaiane-Type Sesquiterpene Dimer from Wik-2 stroemia retusa and its Inhibitory Effects on Histone Acetyl- 3 transferase HBO1 Expression) by Yun et al., could be accepted after covering the following issue.

  • Is the plant reported to have ethnopharmacological uses?
  • Is the plant have a synonyms, if yes please add another plant synonyms in introduction?
  • Discussion of the compound is limited, and author should discuss the NMR in detail.
  • The biosynthetic pathway should be explained in detail with the enzymes incorporated in the process.
  • The part from line 95 t0 101 is repeated, similar to line104 to 110. I think, it is repeated.
  • Why author choose the inhibitory effects on histone acetyl- 3 transferase HBO1 expression, is this due to the type of compounds reported to have this activity? Clarify.
  • In Supplementary, please add more expansion for H NMR in in regions 0.5 to 3.5 and 4.2 to 5 ppm.

Author Response

#2

The manuscript entitled (Retusone A, a Guaiane-Type Sesquiterpene Dimer from Wik-2 stroemia retusa and its Inhibitory Effects on Histone Acetyl- 3 transferase HBO1 Expression) by Yun et al., could be accepted after covering the following issue.

  • Is the plant reported to have ethnopharmacological uses?

Response

There are no known ethnopharmacological uses for this plant.

  • Is the plant have a synonyms, if yes please add another plant synonyms in introduction?

Response

We could not find the synonym of W. retus in literatures.

  • Discussion of the compound is limited, and author should discuss the NMR in detail.

Response

We have further described the HMBC correlations on unit I and unit II in lines 83-90, with accordance in referee’s comments.

  • The biosynthetic pathway should be explained in detail with the enzymes incorporated in the process.

Response

We think that enzymes relate to biosynthetic pathway are very important. However, the enzymes were not identified. Figure 4 is possible biosynthetic pathway.

  • The part from line 95 t0 101 is repeated, similar to line104 to 110. I think, it is repeated.

Response

We have deleted the lines 104-110.

  • Why author choose the inhibitory effects on histone acetyl- 3 transferase HBO1 expression, is this due to the type of compounds reported to have this activity? Clarify.

Response

HBO1 is known to promote cell proliferation in bladder and breast cancers. So, we thought as a strategy that HBO1 expression suppression results in inhibition of MCF7 cells.

  • In Supplementary, please add more expansion for H NMR in in regions 0.5 to 3.5 and 4.2 to 5 ppm.

Response

We have attached enlarge charts of 0.5 to 3.5 and 4.2 to 5 ppm in 1H-NMR spectrum in Supplementary Materials according to referee’s comments.

Reviewer 3 Report

The overall evaluation for the manuscript entitled “Retusone A, a Guaiane-Type Sesquiterpene Dimer from Wik- 2 stroemia retusa and its Inhibitory Effects on Histone Acetyl- 3 transferase HBO1 Expression” is satisfying although it does not appear that the authors have much expertise in the field of chemeprevention usin phytochemical or natural compounds.

The study  is  well organized, the presentation of methods, results and discussion are satisfying.

Observations: I believe it would be better to extend the study also in vivo and in other type of breast cancer cell lines (es. MDA-MB-231, SK-BR-3, etc)

Author Response

#3

The overall evaluation for the manuscript entitled “Retusone A, a Guaiane-Type Sesquiterpene Dimer from Wikstroemia retusa and its Inhibitory Effects on Histone Acetyltransferase HBO1 Expression” is satisfying although it does not appear that the authors have much expertise in the field of chemeprevention usin phytochemical or natural compounds.

The study is well organized, the presentation of methods, results and discussion are satisfying.

Observations: I believe it would be better to extend the study also in vivo and in other type of breast cancer cell lines (es. MDA-MB-231, SK-BR-3, etc)

 Response

We are pleased to note the favorable comments of the referee.

We would like to try to study on normal cells as well as other breast cancer cell lines if we get the more compounds.

Round 2

Reviewer 1 Report

No comment.

Reviewer 2 Report

No comments

Reviewer 3 Report

The overall evaluation for the manuscript “Optimization and validation of a novel three-dimensional co-culture system in decellularized human liver scaffold for the study of liver fibrosis and cancer” is satisfying.

I Believe that the manuscript can ne acceped.